# Analytical Solution for Transient Reactive Elements for DC-DC Converter Circuits

Amr Marey [1],*, Mahajan Sagar Bhaskar [2], Dhafer Almakhles [2] and Hala Mostafa [3]

1 Department of Electrical and Computer Engineering, Faculty of Engineering, University of Alberta Edmonton, Edmonton, AB T6G 1H9, Canada
2 Renewable Energy Laboratory, College of Engineering, Prince Sultan University, Riyadh 11586, Saudi Arabia
3 Department of Information Technology, College of Computer and Information Sciences, Princess Nourah Bint Abdulrahman University, P.O. Box 84428, Riyadh 11671, Saudi Arabia
* Correspondence: amarey@ualberta.ca

**Abstract:** This paper develops an analytical method for modeling the inductor currents and capacitor voltages (ICCV) of a generic DC-DC converter system. The purpose of the designed methodology is to propose a new generalized modeling technique for DC-DC converter systems that accurately models the transient behavior of those systems. The modeled converter is assumed to operate over some number of circuit stages. Each circuit stage can be separately modeled as a linear time-invariant (LTI) system that is solved through the uni-lateral Laplace transform. Furthermore, the initial conditions (ICs) of these LTI systems are related through different algebraic expressions and discrete-time difference equations that originate from the continuity of the ICCV with respect to time. These discrete-time difference equations are then solved with the uni-lateral Z-transform to determine the ICs of the ICCV at each switching period. The generalized theoretical analysis is applied to the study of the transient behavior of the buck-boost converter across various different circuit parameters. This analysis justified with laboratory experimentation of the buck-boost converter, and the transient behavior of the buck-boost converter is compared for each experimental parameter set. The experimental results and the theoretical analysis provide very similar results across the different converter parameters.

**Keywords:** analytical methods; DC-DC converters; power electronics; piece-wise linear differential equations

## 1. Nomenclature and Definitions

Let $\mathbb{R}$ be the set of all real numbers, $\mathbb{C}$ be the set of all complex numbers, $\mathbb{N}_0$ be the set of all non-negative integers, and the set $\mathbb{R}_0^+$ be defined as the set of all non-negative real numbers. Furthermore, let $\mathbb{F}$ denote either $\mathbb{R}$ or $\mathbb{C}$ and $t \in \mathbb{R}$ denote conventional time. $\mathbf{A} \in \mathbb{F}^{r \times m}$ defines a matrix $\mathbf{A}$ with $r$ rows and $m$ columns, such that each element inside this matrix is an element of $\mathbb{F}$; $\mathbf{0}_{r \times m}$ is defined as the additive identity element in $\mathbb{F}^{r \times m}$. Finally, Let $T$ be the matrix transposition operator. The following notations are defined:

If $\mathbf{A} \in \mathbb{F}^{r \times r}$, $e^{\mathbf{A}t}$ is defined as

$$e^{\mathbf{A}t} = \mathbf{I} + \sum_{Q=1}^{\infty} \frac{(\mathbf{A}t)^Q}{Q!}.$$

$\mathbf{I}$ is the multiplicative identity matrix in $\mathbb{F}^{r \times r}$. This infinite series converges for finite $\mathbf{A}$ and $t$ [1].

Consider the column vector $\vec{\mathbf{x}}(\beta) = (x_b(\beta)) = \left[ x_1(\beta) \cdots x_r(\beta) \right]^T \in \mathbb{R}^{r \times 1}$ that is a function of continuous time $\beta \in \mathbb{R}_0^+$ and the column vector $\vec{\mathbf{y}}(\alpha) = (y_b(\alpha)) \in \mathbb{R}^{r \times 1}$ that is

a function of discrete time $\alpha \in \mathbb{N}_0$ for $b = 1, 2 \ldots, r$. The uni-lateral Laplace transform of $\vec{\mathbf{x}}(\beta)$ that maps from $\beta$ to $s \in \mathbb{C}$ is defined as:

$$\mathcal{L}\{\vec{\mathbf{x}}(\beta)\} = \begin{bmatrix} \mathcal{L}\{x_1(\beta)\}, & \cdots & , \mathcal{L}\{x_r(\beta)\} \end{bmatrix}^T \text{ such that}$$
$$\vec{\mathbf{x}}(\beta) = \mathcal{L}^{-1}\{\mathcal{L}\{\vec{\mathbf{x}}(\beta)\}\},$$
$$\vec{\mathbf{x}}(\beta) \in \mathbb{R}^{r \times 1}, \mathcal{L}\{\vec{\mathbf{x}}(\beta)\} \in \mathbb{C}^{r \times 1}.$$

Similarly, the uni-lateral Z-transform of $\vec{\mathbf{y}}(\alpha)$ that maps from $\alpha$ to $z \in \mathbb{C}$ is defined as:

$$\mathcal{Z}\{\vec{\mathbf{y}}(\alpha)\} = \begin{bmatrix} \mathcal{Z}\{y_1(\alpha)\}, & \cdots & , \mathcal{Z}\{y_r(\alpha)\} \end{bmatrix}^T \text{ such that}$$
$$\vec{\mathbf{y}}(\alpha) = \mathcal{Z}^{-1}\{\mathcal{Z}\{\vec{\mathbf{y}}(\alpha)\}\},$$
$$\vec{\mathbf{y}}(\alpha) \in \mathbb{R}^{r \times 1}, \mathcal{Z}\{\vec{\mathbf{y}}(\alpha)\} \in \mathbb{C}^{r \times 1}.$$

## 2. Introduction

Research literature has generated a large influx of new DC-DC converter systems to the market over the past decades. Circuit techniques have also been presented to modify existing DC-DC converters in a more favorable way depending on the application [2]. The literature often focuses on the steady-state (SS) response of the converter. The transient behavior of these converters is often neglected, as the studied converter is assumed to operate in SS for the majority of time [3]. Although this is often a valid assumption for some applications, the SS behavior does not present details on the behavior of the converter in the first few milliseconds of operation. Furthermore, the study of the transient behavior will present the effect of the converter inductances and capacitances on the performance specifications of the converter [4]. As such, this article aims to develop a generalized analytical solution for generic DC-DC converter systems that is capable of accurately modeling the transient response of the converter.

There have been several analytical techniques developed for the modeling of DC-DC converters. One of the earliest analytical techniques developed for DC-DC converter modeling is the the state-space averaging method [5]. First, it is necessary to find the linear state model and the linear switching circuit model of the converter. The next step is the identification of all the state variables. All the models are unified into a single averaged model through a weighted sum [6]. Finally, one may then perform small-signal AC and DC analysis through the linearization and perturbation of the averaged model [7]. Previous works have used state-space averaging to find various information regarding the converter, such as stability and step response [8]. Due to its simplicity and reliability, the state-space averaging technique is still one of the most popular modeling techniques across the literature [9]. State-space averaging is very accurate and has a reasonable simulation time. However, there are several disadvantages of state-space averaging. Firstly, the ripple effects of the inductor currents and capacitor voltages (ICCV) are neglected, as the switching frequency is not taken into account. As such, one can not exploit this technique if high-frequency analysis is required. Furthermore, the model can become very complex if the modeled converter is operating in a discontinuous conduction mode (DCM) or if the modeled converter has a high order [8]. A recent advance in power electronics state-space modeling is presented in [10]. The authors combine Fourier series and classical state-space modeling to model DC-DC with multiple driving pulse-width modulated (PWM) signals without increasing the number of state variables. However, the article neglects the transient behavior of the converter and only focuses on the average SS signal analysis.

Another analytical DC-DC converter circuit modeling technique is the circuit averaging technique [11]. The circuit averaging technique exploits circuit topology. First, the small signal model of the converter is obtained by the averaging of the switch voltages and currents per one switching period. Similar to the state-space averaging method, the linearization and perturbation of the models are applied so the converter transfer functions can be obtained [12]. This modeling technique has been used to find the stability and output impedance of various converters [13]. Unlike the state-space averaging method,

this technique often takes into account parasitic elements and ICCV ripple effects. Furthermore, it is relatively accurate and does not have a long simulation time. Unfortunately, the results developed by this model tend be approximate, as there is a high variation of system parameters oscillating around the DC operating point. This technique can quickly become very chaotic for very high order DC-DC converters. Finally, the technique ignores the high-frequency components of the converter [14].

A new soft-switched inverter circuity and a control mechanism centered around signal flow graph theory is proposed in [15]. Unlike all the other techniques mentioned, the signal flow graph utilizes graphical methods for power electronics' converter analysis. Initially, the voltage and current of each electrical element in the proposed inverter in [15] is assigned a network node in the signal flow graph. Each node is interconnected with the other nodes through the electrical equations governing the elements and the connections amongst the elements. As such, the signal flow graph is generated and Mason's gain equation is applied. The proposed system in [15] is highly efficient due to the fact that the system operates under constant power at the output, and all the switches in the converter have zero-voltage switching capability. This signal flow graph methodology is a very general non-linear analysis methodology that generally has simple mathematics. Furthermore, one can find and assess the response and stability of most DC-DC converters very easily through this method [16]. However, the signal flow graph can become very complex for higher order converters, where there may exist multiple loops inside the graph due to complex circuit topology [17,18].

Refs. [19–21] propose a new modeling technique that accurately models the transient behavior of the classical buck, boost, and buck-boost DC-DC converters. Due to the presence of analog electrical elements such as capacitors and inductors, power electronics systems share some common properties with continuous-time systems. However, due to the inherent switching nature of power electronics' topologies, one can also find that power electronics systems also share some properties with discrete-time systems [22]. As such, refs. [19–21] utilize a combination of both the continuous-time uni-lateral Laplace transform and the uni-lateral discrete-time Z-Transform to model the transient effects of the converter. First, the equations for the ICCV in each circuit stage are determined through the Laplace Transform. Next, the Z-transform and the law of ICCV continuity are used to find the ICCV values at the beginning of each circuit stages. This methodology allows for very detailed analysis of the transient operation of the converter. Furthermore, the method can be utilized to analyze the behavior of the converter at high frequencies. One can use this technique to investigate how inductances, capacitances, load resistances, and PWM duty cycle affect the transient response of a DC-DC converter.

The authors have taken an interest into generalizing the DC-DC converter analytical modeling technique presented in [19–21]. The objective of this paper is to generalize the analytical technique presented in [19–21] to investigate the transient behavior of any DC-DC converter rather than just for the classical buck, boost, and buck-boost DC-DC converters. The DC-DC converter is taken to be as a piece-wise linear system. The Laplace Transform can then be exploited to solve for each linear system in terms of the initial conditions of the ICCV. The initial conditions can be solved by exploiting equations to apply the law of continuity of the ICCV through the Z-transform. The analytical technique applied here can be used to solve for converters operating in either continuous conduction mode (CCM) or DCM. The theoretical analysis of the proposed analytical technique is presented in this paper alongside the relevant hardware-in-loop (HIL) testing in order to validate the presented theory.

Organizing the rest of this work is conducted in the following manner. Section 3 describes the necessary background and general equations from which the proposed modeling technique originates. The solution of these general equations is presented in Section 4. Section 5 specializes the earlier analysis to the buck-boost converter such that the proposed modeling technique can be clarified further to the reader. Section 6 tests the theory presented through hardware-in-loop (HIL) testing of the buck-boost converter. The

effects of inductances, capacitances, load resistances, and PWM duty cycle on the transient behavior of the buck-boost converter are highlighted.

### 3. Problem Description

Let the modeled DC-DC converter have $k$ capacitors, $p$ inductors and $m$ diodes. The input voltage $v_{in}(t)$ is described as the mathematical summation of a DC voltage $V_{DC}$ and an AC voltage ripple $v_{ac}(t)$. Furthermore, there are $n$ circuit stage topologies describing the connections of the electrical circuit components in the converter. For notational simplicity, let $r = k + p$. The state column vector $\vec{x}(t)$ is defined as $\vec{x}(t) = \left[ v_{C_1}, \cdots, v_{C_k}, i_{L_1}, \cdots, i_{L_p} \right]^T$, where $i_{L_q}$ is the current flowing through the $q^{th}$ inductor and $v_{C_w}$ is the voltage across the $w^{th}$ capacitor.

At times $t < 0$, it is assumed that there exists no ICCV. The input column vector $\vec{v}(t)$ is defined as $\vec{v}(t) = \vec{v}_{DC} + \vec{v}_{AC}(t)$, where $\vec{v}_{DC} = \left[ V_{DC}, V_{d_1}, V_{d_2}, \cdots, V_{d_m} \right]^T$ and $\vec{v}_{AC}(t) = \left[ v_{ac}(t), 0, \cdots, \cdots, 0 \right]^T$, where $V_{d_z}$ is the forward voltage drop across the $z^{th}$ diode in the circuit. The switching period of the converter is $T_s$. The $i^{th}$ circuit stage periodically occurs over the half-open interval $\left[ (D_{i-1} + a) T_s, \; (D_i + a) T_s \right)$ for all $a \in \mathbb{N}_0$, and $i = 1, 2, \ldots, n$; such that $D_0 = 0$ and $D_n = 1$. It is assumed that $0 \le D_{i-1} < D_i \le 1$. These assumptions and definitions are accurate for the overwhelming majority of the DC-DC converters.

If each circuit stage topology consists of electrical elements that can each be analytically linearized, which is the case for the majority of DC-DC converters, the ICCV present in the DC-DC converter can be modeled over all time $t$ as

$$\frac{d\vec{x}(t)}{dt} = \sum_{i=1}^{n} \left( \left( \mathbf{A}_i \, \vec{x}(t) + \mathbf{B}_i \, \vec{v}(t) \right) u_i(t) \right). \tag{1}$$

The matrices $\mathbf{A}_i \in \mathbb{R}^{r \times r}$ and $\mathbf{B}_i \in \mathbb{R}^{r \times (m+1)}$ describe the circuit topology from the conventional Kirchhoff's current and voltage laws whenever the converter is operating in the $i^{th}$ circuit stage. The definition of function $u_i(t)$ is

$$u_i(t) = \sum_{\alpha=1}^{\infty} \left( u\left( t - (D_{i-1} + \alpha) T_s \right) - u\left( t - (D_i + \alpha) T_s \right) \right). \tag{2}$$

$u(t)$ is defined as the conventional unit step function that is equal to 1 for all $t \ge 0$ and equal to 0 for all $t < 0$. As such, $u_i(t)$ can be considered as a pulse train that is equal to 1 for all $t$, where the circuit is operating in the $i^{th}$ circuit stage and 0 otherwise. A graphical representation of $u_i(t)$ is presented in Figure 1. Finally, although $\vec{x}(t)$ is presented in this paper to model the main inductances and capacitances, one may also include stray inductances and capacitances inside $\vec{x}(t)$.

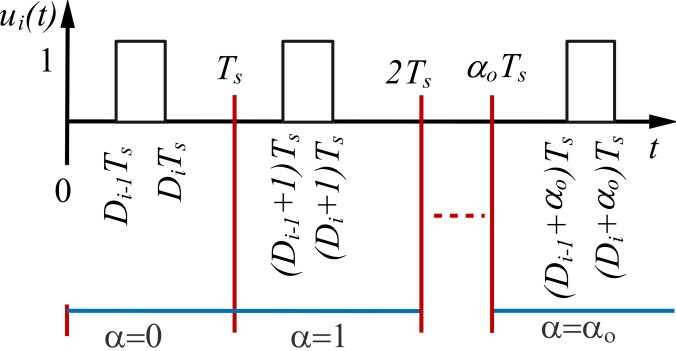

**Figure 1.** Graphical representation of periodic pulse $u_i(t)$.

## 4. Theoretical Analysis

### 4.1. Deriving the Analytical Solution

Due to the heavily discontinuous nature of (1) caused by the switching devices found in the DC-DC converter circuitry, one cannot use the conventional techniques that are usually used to solve continuous time ordinary linear differential equations. As such, consider the variable substitution

$$t = (\alpha + \beta)T_s \ \text{ where } \ \alpha = floor(t) \text{ and } \ \beta = \frac{t}{T_s} - \alpha. \tag{3}$$

At any time $t \in \mathbb{R}_0^+$, $\alpha$ represents the total integer number of switching periods that have passed since initial time $t = 0$. $\beta$ represents the percentage of completion of the current switching period. Through (3), one can show $\alpha \in \mathbb{N}_0$ and $\beta \in [0, 1)$ for all $t \in \mathbb{R}_0^+$.

From (2) and Figure 1, one can observe that the value of $u_i(t)$ is completely independent of the number of switching periods completed $\alpha$ and only dependent on the percentage of completion of the current switching period $\beta$. As such one may re-express $u_i(t)$ as

$$u_i(t) = u_i(\beta) = \left\{ \begin{array}{ll} 1 & D_{i-1} \leq \beta < D_i \\ 0 & \text{otherwise} \end{array} \right. . \tag{4}$$

Vectors $\vec{\mathbf{x}}(t)$ and $\vec{\mathbf{v}}(t)$ can now be described as functions of $\alpha$ and $\beta$ such that $\vec{\mathbf{x}}(t) = \vec{\mathbf{x}}(\alpha, \beta)$ and $\vec{\mathbf{v}}(t) = \vec{\mathbf{v}}(\alpha, \beta)$. Furthermore, for any constant positive integer $\alpha = \alpha_0$, one can use (3) to show

$$\frac{\Delta t}{\Delta \beta} = T_s \ \text{ such that } \ \lim_{\Delta \beta \to 0} \frac{\Delta t}{\Delta \beta} = \frac{dt}{d\beta} = T_s \text{ for } \alpha = \alpha_0. \tag{5}$$

One can now substitute (5) into (1) such that $n$ vector differential equations may be written to model the DC-DC circuitry over the $\alpha_0$−th switching period

$$\frac{d\vec{\mathbf{x}}(\alpha_0, \beta)}{d\beta} = T_s \left( \mathbf{A}_i \, \vec{\mathbf{x}}(\alpha_0, \beta) + \mathbf{B}_i \, \vec{\mathbf{v}}(\alpha_0, \beta) \right)$$
$$\text{for } D_{i-1} \leq \beta < D_i \quad \text{where } i = 1, 2, \cdots, n. \tag{6}$$

$\mathbf{A}_i$ and $\mathbf{B}_i$ are constant over the interval $\beta \in [D_{i-1}, D_i)$, as it was mentioned earlier that each of those matrices is constant over each circuit stage. Each equation listed in the family of $n$ Equation (6) is a one-dimensional differential equation with independent continuous time variable $\beta$. $\alpha = \alpha_0$ does not vary, as it was assumed to be constant by definition. The uni-lateral Laplace transform that maps from $\beta$ to the variable $s \in \mathbb{C}$ can be used to provide a solution for the $i$−th equation presented in the family of Equation (6). Let $\mathbf{E}_i = \left( s\mathbf{I} - T_s \mathbf{A}_i \right)^{-1}$. After a straight forward computation, the solution for the $i$−th equation presented in (6) can be demonstrated to be [23]

$$\vec{\mathbf{x}}(\alpha_0, \beta) = \vec{\mathbf{x}}_{SS-i}(\alpha_0, \beta) + \vec{\mathbf{x}}_{IC-i}(\alpha_0, \beta) \text{ for } D_{i-1} \leq \beta < D_i, \tag{7}$$

$$\vec{\mathbf{x}}_{SS-i}(\alpha_0, \beta) = T_s \, \mathcal{L}^{-1} \left\{ e^{-D_{i-1}s} \mathbf{E}_i \left( \mathbf{B}_i \, \mathcal{L}\{\vec{\mathbf{v}}(t + D_{i-1})\} \right) \right\}$$
$$= T_s \int_{D_{i-1}}^{\beta} e^{(T_s \mathbf{A}_i(\beta - D_{i-1} - \tau))} \, \mathbf{B}_i \, \vec{\mathbf{v}}(\alpha_0, \tau) d\tau, \tag{8}$$

$$\vec{\mathbf{x}}_{IC-i}(\alpha_0, \beta) = \mathcal{L}^{-1} \left\{ e^{-D_{i-1}s} \mathbf{E}_i \, \vec{\mathbf{x}}(\alpha_0, D_{i-1}) \right\}$$
$$= e^{T_s \mathbf{A}_i(\beta - D_{i-1})} \, \vec{\mathbf{x}}(\alpha_0, D_{i-1}), \tag{9}$$

where, $\vec{\mathbf{x}}_{SS-i}(\alpha_0, \beta)$ models the steady-state response to (6) and $\vec{\mathbf{x}}_{IC-i}(\alpha_0, \beta)$ models the transient response to (6). $\mathcal{L}\{\vec{\mathbf{v}}(t)\}$ can be further simplified as,

$$\mathcal{L}\{\vec{\mathbf{v}}(t)\} = \frac{1}{s} \vec{\mathbf{v}}_{DC} + \mathcal{L}\{\vec{\mathbf{v}}_{AC}(t)\}. \tag{10}$$

Since all the elements inside the vector $\vec{\mathbf{x}}(\alpha_0, \beta)$ represent either a capacitor voltage or an inductor current; therefore, $\vec{\mathbf{x}}(t)$ is continuous for all time $t$ despite any spontaneous change in the circuit topology generated from switch action. As such, at time $t = \alpha_0 + D_i$, when the circuit topology is changed from the $i-$th circuit stage of the $\alpha_0$-th switching period to the $(i+1)-$th circuit stage of the $\alpha_0-$th switching period, the vector $\vec{\mathbf{x}}(t)$ must maintain continuity with respect to time. This can be presented in the $(\alpha, \beta)$ domain as

$$\lim_{(\alpha,\beta)\to(\alpha_0, D_i^-)} \vec{\mathbf{x}}(\alpha, \beta) = \lim_{(\alpha,\beta)\to(\alpha_0, D_i^+)} \vec{\mathbf{x}}(\alpha, \beta) \tag{11}$$

$$\forall i = 1, 2, \ldots, n-1.$$

Applying (11) yields

$$\vec{\mathbf{x}}(\alpha_0, D_i) = \vec{\mathbf{x}}_{SS-i}(\alpha_0, D_i^-) + e^{T_s \mathbf{A}_i(D_i - D_{i-1})} \vec{\mathbf{x}}(\alpha_0, D_{i-1}). \tag{12}$$

(12) can be used repeatedly to derive a mathematical function between the initial conditions of the $i-$th circuit stage of the $\alpha_0-$th switching period ($\vec{\mathbf{x}}(\alpha_0, D_{n-1})$) in terms of the 1-st circuit stage of the $\alpha_0-$th switching period ($\vec{\mathbf{x}}(\alpha_0, 0)$). As such, $\vec{\mathbf{x}}(\alpha_0, D_i)$ can be expressed as $\vec{\mathbf{x}}(\alpha_0, D_i) = f_i(\vec{\mathbf{x}}(\alpha_0, 0))$. Furthermore, the continuity of the vector $\vec{\mathbf{x}}(t)$ or $\vec{\mathbf{x}}(\alpha, \beta)$ with respect to time must also be exhibited as soon as the circuit is changed from the $n-$th circuit stage of the $\alpha$-th switching cycle to the 1-st circuit stage of the $(\alpha + 1)-$th switching cycle. This can be presented as

$$\lim_{(\alpha,\beta)\to(\alpha, 1^-)} \vec{\mathbf{x}}(\alpha, \beta) = \lim_{(\alpha,\beta)\to(\alpha+1, 0^+)} \vec{\mathbf{x}}(\alpha, \beta). \tag{13}$$

By placing Equations (8), (9) and (12) into (13), the following is obtained:

$$\begin{aligned} \vec{\mathbf{x}}(\alpha + 1, 0) &= \vec{\mathbf{g}}_{SS}(\alpha) + \vec{\mathbf{g}}_{IC}(\alpha) \text{ such that} \\ \vec{\mathbf{g}}_{SS}(\alpha) &= \vec{\mathbf{x}}_{SS-n}(\alpha, 1^-) \\ \vec{\mathbf{g}}_{IC}(\alpha) &= e^{T_s \mathbf{A}_n(1^-)} f_{n-1}(\vec{\mathbf{x}}(\alpha, 0)). \end{aligned} \tag{14}$$

One can note that (14) is effectively a linear discrete time difference equation, as it varies over the non-negative integer $\alpha$ only. As such, one may use the uni-lateral Z-transform that maps from $\alpha_0 \in \mathbb{N}_0$ to $z \in \mathbb{C}$. The solution to (13) is easily shown to be [24]

$$\vec{\mathbf{x}}(\alpha, 0) = \mathcal{Z}^{-1}\left\{ \frac{1}{z}\left( \mathcal{Z}\{\vec{\mathbf{g}}_{SS}(\alpha) + \vec{\mathbf{g}}_{IC}(\alpha)\} + \vec{\mathbf{x}}(0, 0) \right) \right\}. \tag{15}$$

Observe that $\vec{\mathbf{x}}(0, 0) = \vec{\mathbf{x}}(t)\Big|_{t=0}$. The average of $\vec{\mathbf{x}}(\alpha, \beta)$ over one switching period, $\vec{\mathbf{x}}_{avg}(\alpha, \beta)$ can be obtained through

$$\vec{\mathbf{x}}_{avg}(\alpha, \beta) = \int_0^1 \vec{\mathbf{x}}(\alpha, \beta) d\beta. \tag{16}$$

By taking the Z-transform of both sides in (16), one can obtain the transfer function of each component inside vector $\vec{\mathbf{x}}(\alpha, \beta)$ with respect to the input voltage $v_{in}(\alpha, \beta)$. Furthermore, to obtain the average steady-state value of $\vec{\mathbf{x}}(\alpha, \beta)$, which is denoted as $\vec{\mathbf{x}}_{avg-SS}$, the following equation can be applied:

$$\vec{\mathbf{x}}_{avg-SS} = \lim_{\alpha\to\infty}\left( \int_0^1 \vec{\mathbf{x}}(\alpha, \beta) d\beta \right). \tag{17}$$

With the assumption that the initial condition of $\vec{\mathbf{x}}(t)$ at $t = 0$ is given, and the ability to inter-relate the initial conditions of each circuit stage over the same switching period from (12), and the ability to interrelate the the initial conditions of two consecutive switching

periods; one can analytically model $\vec{\mathbf{x}}(t)$ for all time. This allows for one to obtain the transient ICCV response with precise accuracy. This analysis was conducted solely through piecewise-linear methods. The mathematical methods applied throughout this modeling technique are summarized in the flowchart presented in Figure 2.

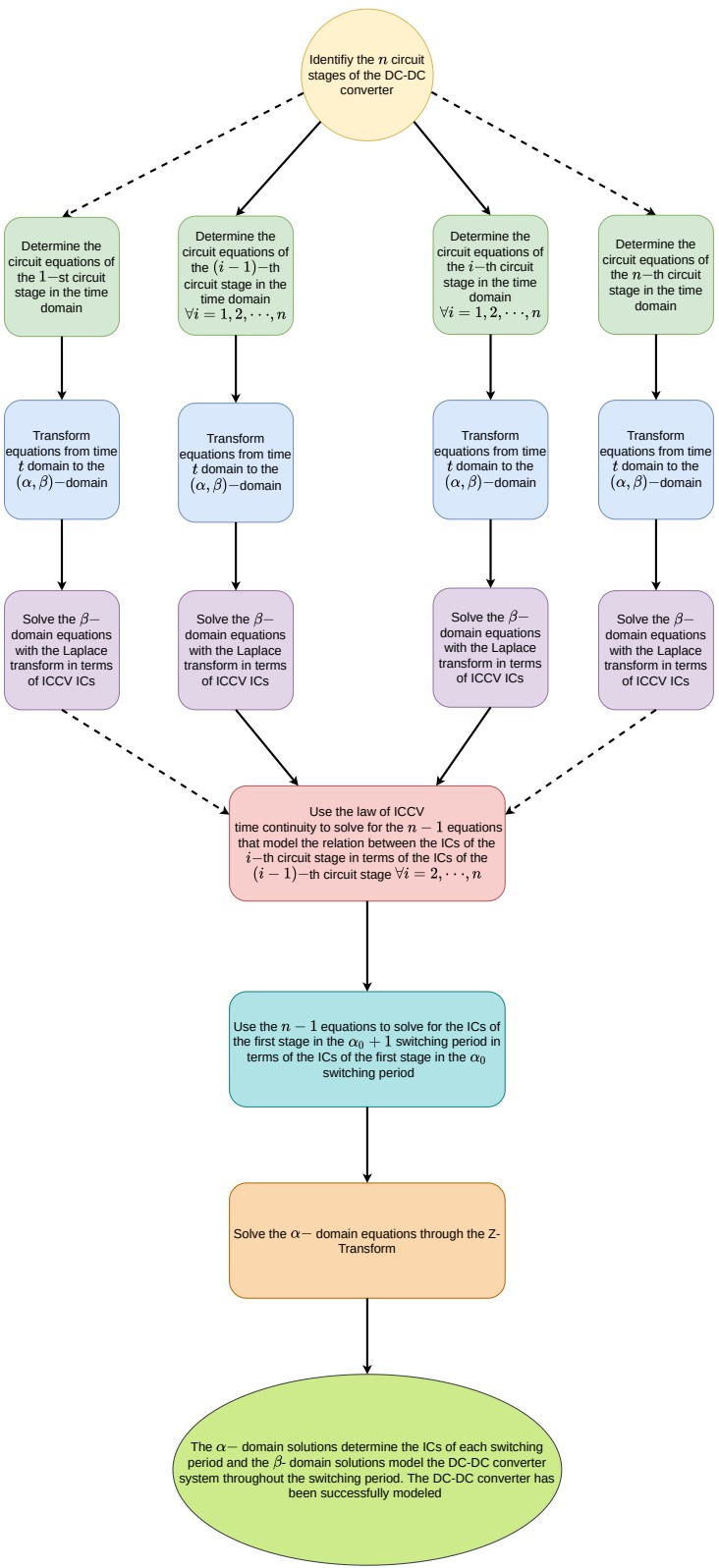

**Figure 2.** Brief summary of the proposed modeling method for DC-DC converter systems.

### 4.2. Comparison of Proposed Method with Other Modeling Techniques

The proposed modeling technique focuses very heavily on the transient effects of DC-DC converter systems and on the parasitic electrical elements. Table 1 compares the advantages and disadvantages to the proposed method to the most popular power electronics modeling techniques presented in the literature. In general, this technique is very advantageous for power electronics control systems applications, where the transient behavior of the converter is absolutely crucial. Through this technique, we can easily identify various transient parameters such as peak overshoot, settling time, etc. This will be shown in Section 6. However, a main disadvantage of this technique is that the equations can become very chaotic depending on the variety of parasitic elements being modeled.

**Table 1.** Summary of advantages and disadvantages of different power electronics DC-DC converter systems across the literature.

| | Advantages | Disadvantages |
|---|---|---|
| State-Space Averaging | • The averaged equations are derived directly from the circuit topology.<br>• The analysis presented is well-defined and mathematically rigorous.<br>• Reasonable accuracy is provided with quick simulation time.<br>• It is still a very popular modeling technique amongst the literature. | • The high frequency components are often neglected and as such it cannot be used for high-frequency analysis.<br>• The ripple effects of the ICCV are ignored, as the switching frequency is neglected.<br>• The modeling equations become very complex if the converter is operating in DCM or if the modeled converter has a complex circuit topology. |
| Circuit Averaging | • ICCV ripple and parasitic effects are often taken into account.<br>• The analysis is even simpler than that of state-space averaging.<br>• It is still a common modeling technique despite being one of the earliest modeling techniques for power electronics systems. | • The high frequency components are often neglected and as such it cannot be used for high-frequency analysis.<br>• The modeling equations become very complex if the converter is operating in DCM or if the modeled converter has a complex circuit topology.<br>• The final model has some loss of accuracy due to the variation of system parameters around the DC operating point. |
| Signal Flow Graph | • It is a fast and quick analysis technique that does not need advanced mathematics.<br>• The characteristics polynomial of the circuit is derived very quickly. This can be used to calculate the response and stability of the DC-DC converter.<br>• It is a general method that can easily be applied across different DC-DC converters. | • In complex circuit topologies, multiple loops may exist. This makes the determination of stability and network compensation design difficult. |
| Proposed Method | • The transient dynamics of the converter are modeled with very high accuracy.<br>• The technique is easily generalized and can be applied to a large variety of DC-DC converters.<br>• This technique can be applied for high frequency analysis.<br>• The ICCV ripple effects are clearly modeled to enhance accuracy. | • The modeling equations can become long and complicated. |

## 5. Example: Transient Analysis of the Buck-Boost Converter

The buck-boost converter, presented in Figure 3, is one of the most iconic DC-DC converters due to its ability to either step-up or step-down an input voltage. Furthermore, its simple circuit topology makes it very applicable for a large number of industries and allows for relatively simple mathematical analysis regarding its operation. As such, it is chosen as the example that clarifies the theory presented in this paper.

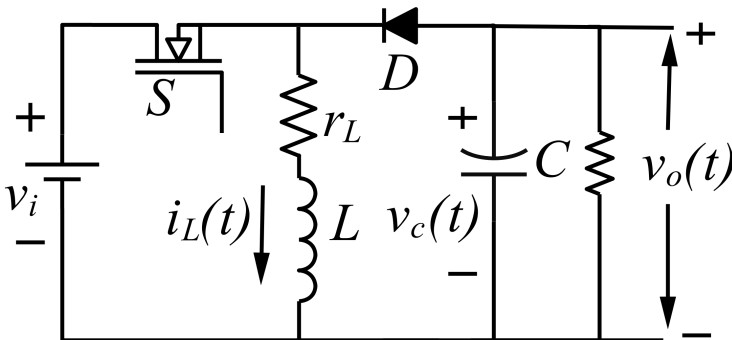

**Figure 3.** Buck-boost converter circuitry.

The buck-boost converter has two operational modes in CCM ($n = 2$). The operational modes are presented in Figure 4. Only the inductor $L$ with equivalent series resistance $r_L$ is treated as a non-ideal electrical element due to the fact that the non-ideality of the inductors is usually a large portion of all the non-idealities found in DC-DC converters. The rest of the components are treated as ideal to simplify the mathematical analysis presented in this paper and to highlight inductor effects on the transient behavior of the buck-boost converter. Regarding the buck-boost circuit topology, the state column vector is $\vec{x}(t) = \begin{bmatrix} i_L(t), & v_C(t) \end{bmatrix}^T$ and the input column vector is $\vec{v}(t) = \begin{bmatrix} v_{in}(t), & 0 \end{bmatrix}^T$. Throughout this example, it will be assumed that $\vec{x}(t = 0) = \mathbf{0}_{2 \times 1}$ such that there is no initial current or voltage flowing across any of the circuit components. The switch in the buck-boost converter is operated a duty cycle $D$. The matrices $\mathbf{A}_1, \mathbf{A}_2, \mathbf{B}_1,$ and $\mathbf{B}_2$, which describe the buck-boost topology, are

$$\mathbf{A}_1 = \begin{bmatrix} -r_L L^{-1} & 0 \\ 0 & -(RC)^{-1} \end{bmatrix}, \mathbf{B}_1 = \begin{bmatrix} L^{-1} & 0 \\ 0 & 0 \end{bmatrix}, \tag{18}$$

$$\mathbf{A}_2 = \begin{bmatrix} -r_L L^{-1} & L^{-1} \\ -C^{-1} & (RC)^{-1} \end{bmatrix}, \mathbf{B}_2 = \begin{bmatrix} 0 & -L^{-1} \\ 0 & 0 \end{bmatrix}. \tag{19}$$

The input voltage source ripple is neglected such that $v_{in}(t)$ is equal to DC voltage $V_i$.

As demonstrated previously, attempting to solve (1) requires solving the group of systems of differential Equation (6). For $\beta \in [0, D)$ and a constant $\alpha$ ($\alpha = \alpha_0$), the current flowing through inductor $L$ and the output voltage across capacitor $C$ are

$$i_L(\alpha_0, \beta) = \frac{V_i}{r_L} + \left(i_L(\alpha_0, 0) - \frac{V_i}{r_L}\right)e^{\frac{-\beta T r_L}{L}} \tag{20}$$

$$v_C(\alpha_0, \beta) = -v_o(\alpha_0, 0)e^{\frac{-\beta T_s}{RC}} \tag{21}$$

Over the interval $\beta \in [D, 1)$ and $\alpha = \alpha_0$, $i_L(\alpha_0, \beta)$ and $v_C(\alpha_0, \beta)$ are

$$i_L(\alpha_0, \beta) = e^{-k_1(\beta - D)} \Big( i_L(\alpha_0, D) \cos\Big(k_2(\beta - D)\Big) +$$
$$\Big(k_3 i_L(\alpha_0, D) - k_4 v_C(\alpha_0, D)\Big) \sin\Big(k_2(\beta - D)\Big)\Big), \tag{22}$$

$$v_C(\alpha_0, \beta) = -e^{-k_1(\beta-D)}\Big(v_C(\alpha_0, D)\cos\big(k_2(\beta-D)\big) + \big(k_5 i_L(\alpha_0, D) - k_3 v_C(\alpha_0, D)\big)\sin\big(k_2(\beta-D)\big)\Big),$$ (23)

where

$$k_1 = \left(\frac{(L + r_L RC)T_s}{2RLC}\right)$$ (24a)

$$k_2 = \sqrt{\frac{T_s^2 r_L + R(T_s^2 - LCk_1^2)}{LC}}$$ (24b)

$$k_3 = \frac{(L - r_L RC)T_s}{2RLCk_2}$$ (24c)

$$k_4 = \frac{T_s}{k_2 L}$$ (24d)

$$k_5 = \frac{T_s}{k_2 C}$$ (24e)

Equations (20)–(23) are the buck-boost topology localized versions of (7); the four equations came from solving (6), (18) and (19). Now, using (11), (20) and (22) to determine $i_L(\alpha_0, D)$ in terms of $i_L(\alpha_0, 0)$ yields

$$i_L(\alpha_0, D) = \left(i_L(\alpha_0, 0) - \frac{V_i}{r_L}\right)e^{\frac{-DT_s r_L}{L}} + \frac{V_i}{r_L}.$$ (25)

Similarly, using (11), (21) and (23) to determine $v_C(\alpha_0, D)$ in terms of $v_C(\alpha_0, 0)$ yields

$$v_C(\alpha_0, D) = -v_o(\alpha_0, 0)e^{\frac{-DT_s}{RC}}.$$ (26)

Now, (14) is applied to determine the relationship between $\vec{x}(\alpha, 0)$ and $\vec{x}(\alpha + 1, 0)$. As such,

$$i_L(\alpha + 1, 0) = e^{-k_1(1-D)}\Big(i_L(\alpha, D)\cos\big(k_2(1-D)\big) + \big(k_3 i_L(\alpha, D) - k_4 v_C(\alpha, D)\big)\sin\big(k_2(1-D)\big)\Big),$$ (27)

$$v_C(\alpha + 1, 0) = -e^{-k_1(1-D)}\Big(v_C(\alpha, D)\cos\big(k_2(1-D)\big) + \big(k_5 i_L(\alpha, D) - k_3 v_C(\alpha, D)\big)\sin\big(k_2(1-D)\big)\Big).$$ (28)

(27) and (28) constitute a system of linear discrete-time difference equations. The Z-transform is used to solve this system to show

$$i_L(\alpha, 0) = \mathcal{Z}^{-1}\Big\{\frac{r_L^{-1}z\big((h_4 + h_5)z + h_6 + h_7\big)}{(z-1)\big(z^2 + (h_0 + h_1)z + h_2 + h_3\big)}\Big\}$$ (29)

$$v_C(\alpha, 0) = \mathcal{Z}^{-1}\Big\{\frac{-r_L^{-1}h_8 z}{(z-1)\big(z^2 + (h_0 + h_1)z + h_2 + h_3\big)}\Big\}$$ (30)

where,

$$h_0 = -e^{k_1(D-1)}\left(e^{\frac{-DT_s}{RC}} + e^{\frac{-DT_s r_L}{L}}\right)\cos\left(k_2(D-1)\right) \tag{31a}$$

$$h_1 = k_3 z e^{k_1(D-1)}\left(e^{\frac{-DT_s r_L}{L}} - e^{\frac{-DT_s}{RC}}\right)\sin\left(k_2(D-1)\right) \tag{31b}$$

$$h_2 = e^{\frac{-DT_s r_L}{L}} e^{\frac{-DT_s}{RC}} e^{2k_1(D-1)}\cos^2\left(k_2(D-1)\right) \tag{31c}$$

$$h_3 = e^{\frac{-DT_s r_L}{L}} e^{\frac{-DT_s}{RC}} e^{2k_1(D-1)} K \sin^2\left(k_2(D-1)\right) \tag{31d}$$

$$h_4 = -V_i e^{k_1(D-1)}(1 - e^{\frac{-DT_s r_L}{L}})\cos\left(k_2(D-1)\right) \tag{31e}$$

$$h_5 = V_i k_3 e^{k_1(D-1)}(1 - e^{\frac{-DT_s r_L}{L}})\sin\left(k_2(D-1)\right) \tag{31f}$$

$$h_6 = V_i e^{\frac{-DT_s}{RC}} e^{2k_1(D-1)}(1 - e^{\frac{-DT_s r_L}{L}})\cos^2\left(k_2(D-1)\right) \tag{31g}$$

$$h_7 = V_i e^{\frac{-DT_s}{RC}} e^{2k_1(D-1)}(1 - e^{\frac{-DT_s r_L}{L}}) K \sin^2\left(k_2(D-1)\right) \tag{31h}$$

$$h_8 = V_i k_5 e^{k_1(D-1)}(1 - e^{\frac{-DT_s r_L}{L}})\sin\left(k_2(D-1)\right) \tag{31i}$$

where $K = (k_4 k_5 - k_3^2)$. To simplify further analysis, the following constants are defined

$$g_0 = h_0 + h_1 \tag{32a}$$

$$g_1 = h_2 + h_3 \tag{32b}$$

$$g_2 = -r_L^{-1} \tag{32c}$$

$$g_3 = h_4 + h_5 \tag{32d}$$

$$g_4 = h_6 + h_7 \tag{32e}$$

$$g_5 = -r_L^{-1} h_8 \tag{32f}$$

The analytical expression of the inverse z-transforms in (29) and (30) exists and has been computed by the authors. For all positive integers $\alpha$, the values of $i_L$ and $v_C$ at the beginning of every switching cycle is

$$i_L(\alpha, 0) = \frac{a_0 + a_1(-1)^\alpha g_1^{\frac{\alpha}{2}} \cos\left(\alpha \arccos(\frac{g_0}{2\sqrt{g_1}})\right)}{g_0(g_0 + g_1 + 1)} \tag{33}$$

$$v_C(\alpha, 0) = \frac{g_0 g_5 + 2g_5(-1)^\alpha g_1^{\frac{\alpha}{2}} \cos\left(\alpha \arccos(\frac{g_0}{2\sqrt{g_1}})\right)}{g_0(g_0 + g_1 + 1)} \tag{34}$$

where

$$a_0 = g_0 g_2(g_3 + g_4) \tag{35a}$$

$$a_1 = -2g_2(g_0 g_3 - g_4 + g_1 g_3) \tag{35b}$$

As such, through (33), (20), (22), (34), (21) and (23) one is able to determine the transient behavior $i_L(t)$ and $v_C(t)$ through the substitutions $\alpha = floor(t)$ and $\beta = tT_s^{-1} - \alpha$ as shown in (3). These equations allow us to explore the effects of inductances and capacitances on the transient behavior of DC-DC converter systems in a manner that is mathematically rigorous.

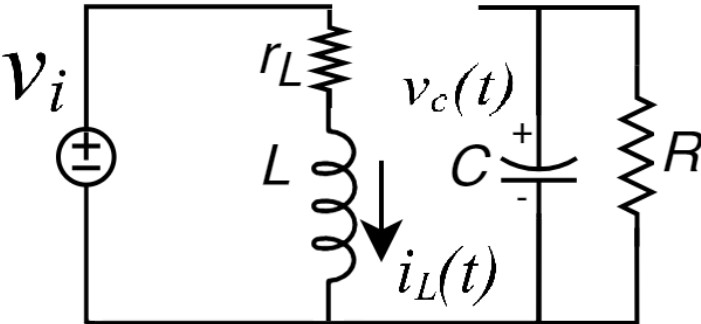

(**a**) Buck-boost circuit topology during the ON state.

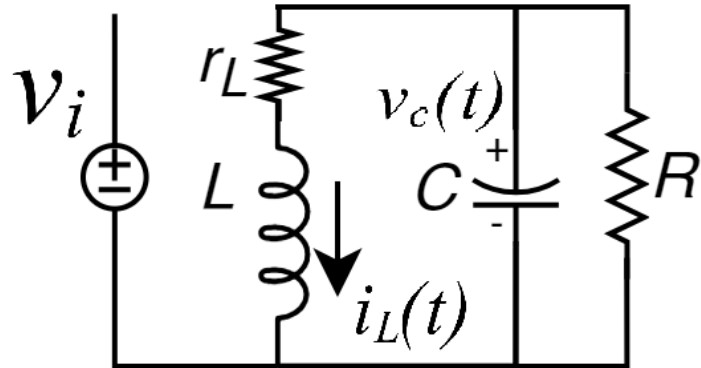

(**b**) Buck-boost circuit topology during the OFF state.

**Figure 4.** Buck-boost converter circuit stages, (**a**) On state; (**b**) OFF state.

### 6. Experimental Justification of the Theory

The theory presented in this paper is experimentally justified through the observation of a buck-boost converter tested by power electronics hardware-in-loop (HIL) FPGA experimentation. Four different parameter sets of the buck-boost circuit are chosen to demonstrate the effects of the passive elements and their non-idealities on the transient response of the ICCV of the buck-boost converter. The results observed through a digital oscilloscope are compared to the theoretical results that would have been obtained through the graphing of the analytical equations presented in this paper. In all the four parameter sets, the values of $T_s$, $D$, , $V_i$ are $T_s = \frac{1}{5000}s$, $D = 0.75$, $V_i = 24$ V. In the first parameter set, parameter set $S_1$, the values of the circuit components used are $R = 60$ $\Omega$, $L = 5 \times 10^{-3}$ H, $r_L = 0.8$ $\Omega$, $C = 220 \times 10^{-6}$ F. In parameter set $S_2$, the values of $R$, $L$, $r_L$ are unchanged from parameter set $S_1$ but the capacitance $C$ is $C = 120 \times 10^{-6}$ F. In parameter set $S_3$, only the values of $L$ and $r_L$ are changed from $S_1$ to $L = 9 \times 10^{-3}$ H, and $r_L = 1.2$ $\Omega$. In the final parameter set $S_4$, only the value of the load resistance $R$ is changed from $S_1$ to $R = 100$ $\Omega$.

To yield the graph of the theoretical inductor current and capacitor voltage at the beginning of each switching period for each parameter set, (29) and (30) are applied for each parameter set. The theoretical initial conditions at each switching period are presented in Figures 5–8. These initial conditions allow for the graphical expression of $i_L$ and $v_C$ over continuous time through (20)–(23). These graphs are shown in Figures 9–12. Next, the results are obtained experimentally from HIL to compare the theoretical model presented in this paper with the experimental evidence. The experimental evidence from the HIL is shown in Figures 13–16. As one can observe, the experimental ICCV seems to be identical to the theoretical ICCV for each parameter set. As one increases the output capacitance in the buck-boost converter, the time-duration it takes to reach steady-state increases, as shown from Figures 9, 10, 13 and 14. Furthermore, this increase in capacitance decreases the maximum current flowing through the inductor and increases the maximum voltage across the capacitor. An increase in the inductance seems to also increase the time-duration

it takes to reach steady-state increases, as shown from Figures 9, 11, 13 and 15; the increase in $r_L$ has a strong effect on the steady-state ICCV, as they decreased in $S_3$ compared to $S_1$. Finally, an increase in the resistance $R$ seems to have a similar effect to that of an increase in capacitance or inductance regarding the time-duration it takes to reach steady-state.

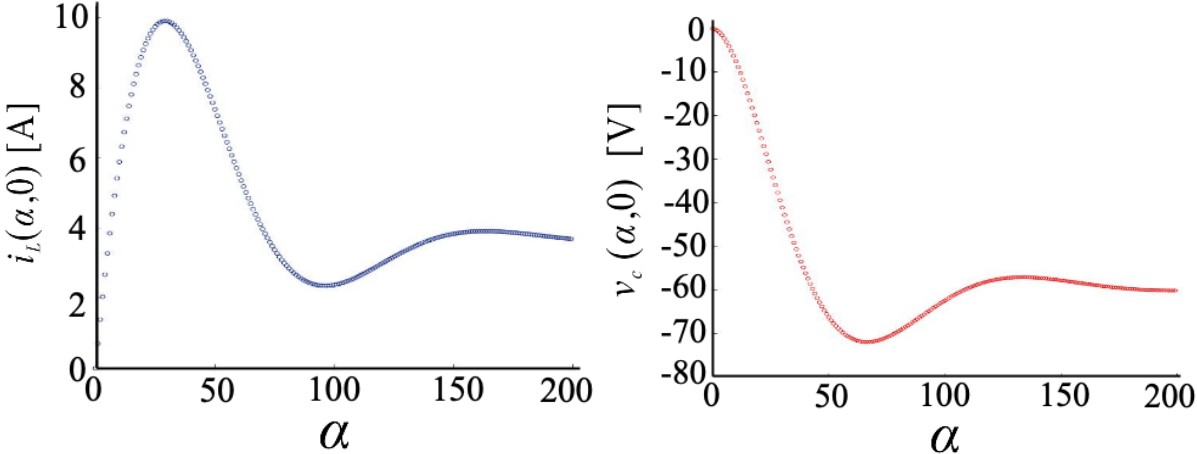

**Figure 5.** Graph of ICCV at the beginning of the $\alpha$-th switching period from $S_1$ parameters through Equations (33) and (34).

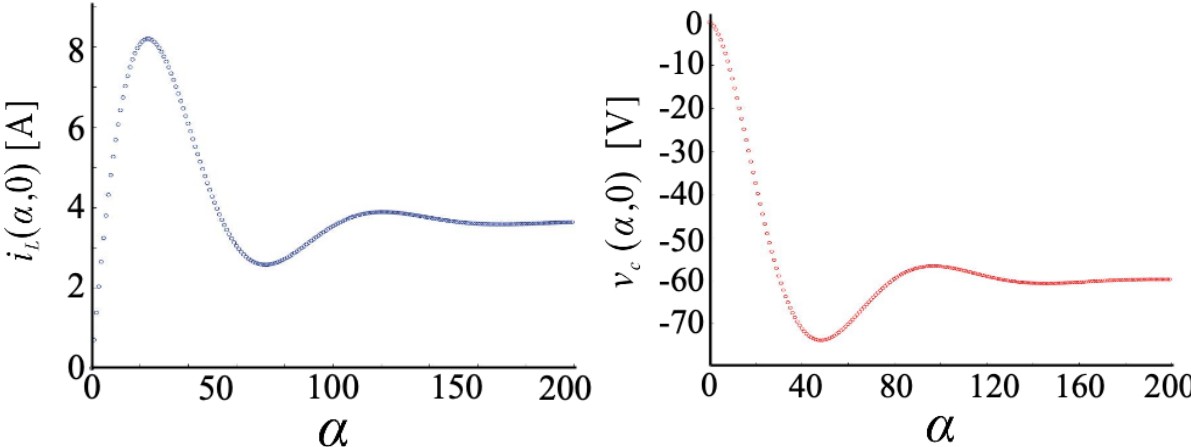

**Figure 6.** Graph of the ICCV at the beginning the $\alpha$-th switching period from $S_2$ parameters through analytical Equations (33) and (34).

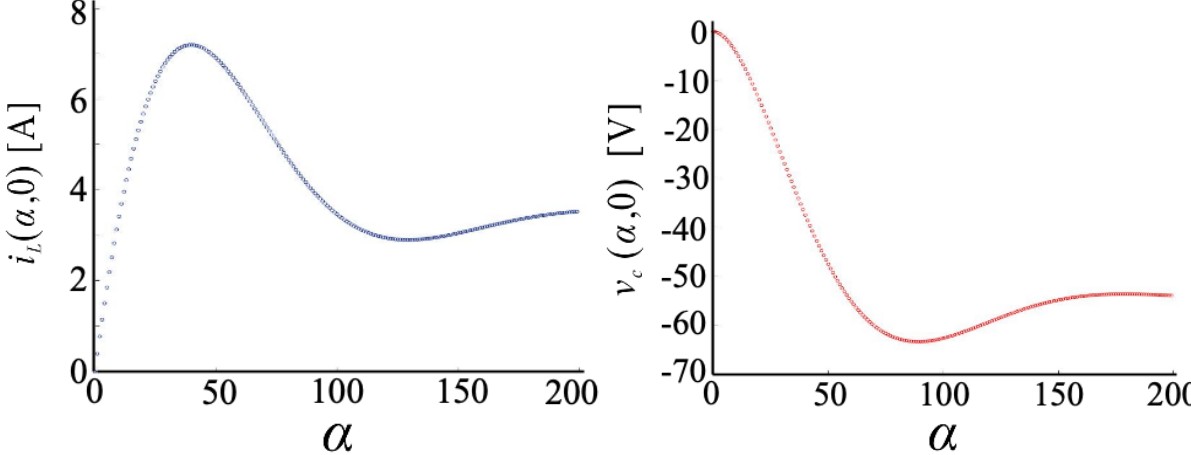

**Figure 7.** Graph of the ICCV at the beginning of the $\alpha$-th switching period from $S_3$. parameters through analytical Equations (33) and (34).

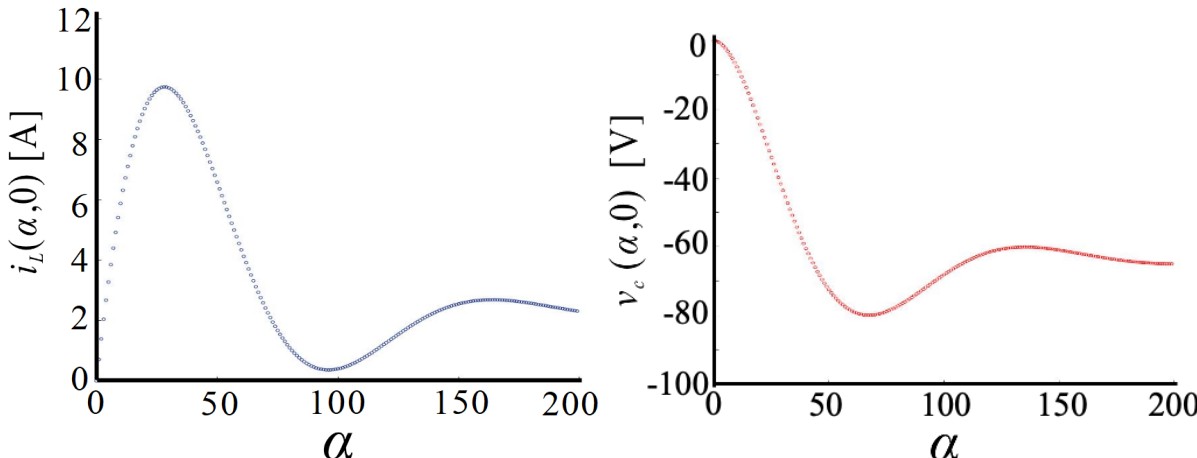

**Figure 8.** Graph of the ICCV at the beginning of the $\alpha$-th switching period from $S_4$ parameters through analytical Equations (33) and (34).

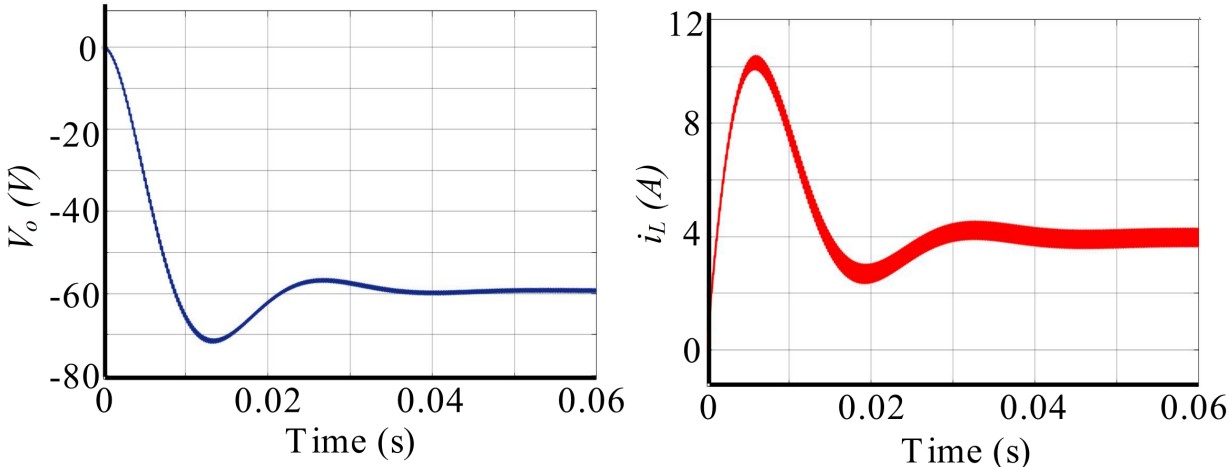

**Figure 9.** Graph of the $S_1$ parameter set ICCV in continuous time through (20)–(23) and knowledge of the ICCV at the beginning of each switching period.

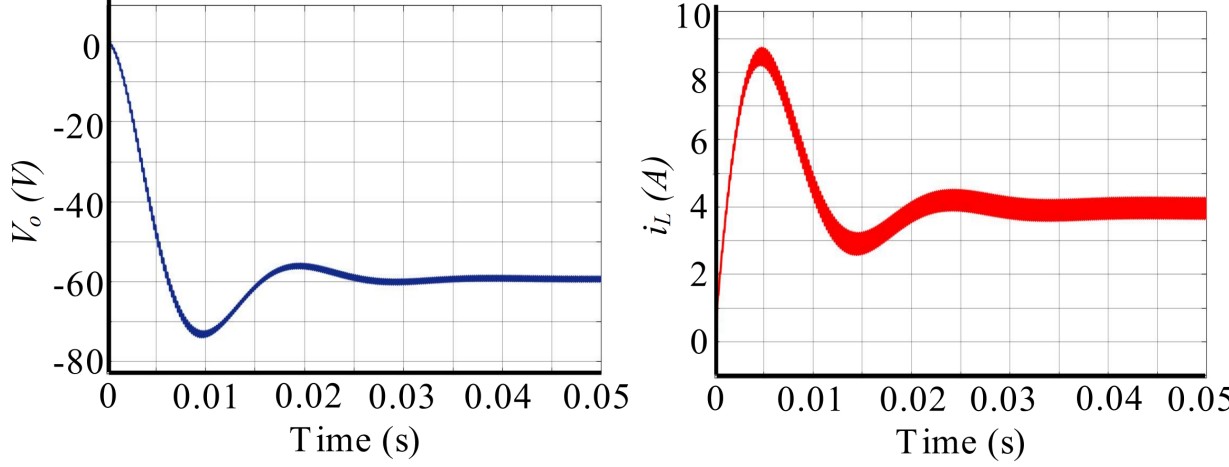

**Figure 10.** Graph of the $S_2$ parameter set ICCV in continuous time through (20)–(23) and knowledge of the ICCV at the beginning of each switching period.

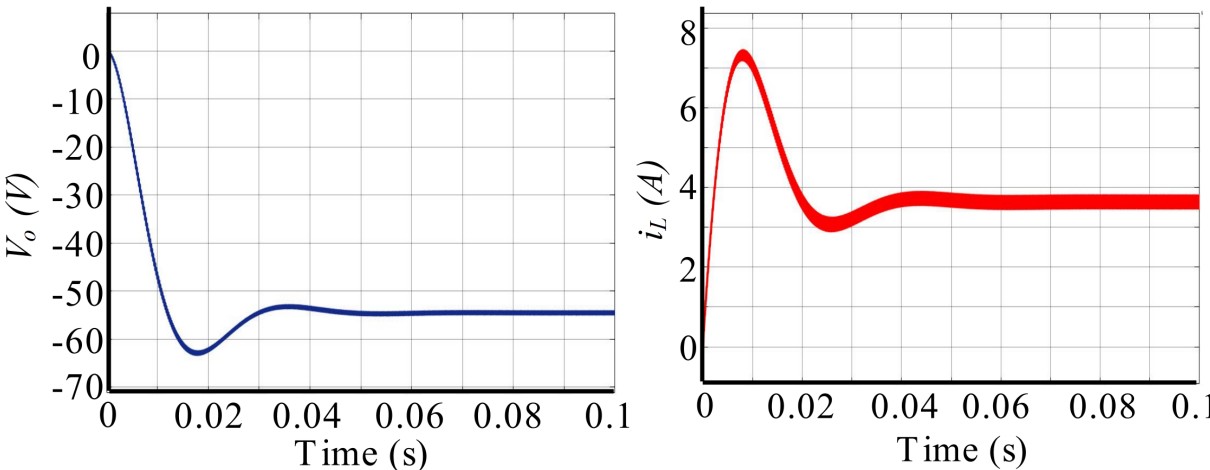

**Figure 11.** Graph of the $S_3$ parameter set ICCV in continuous time through (20)–(23) and knowledge of the ICCV at the beginning of each switching period.

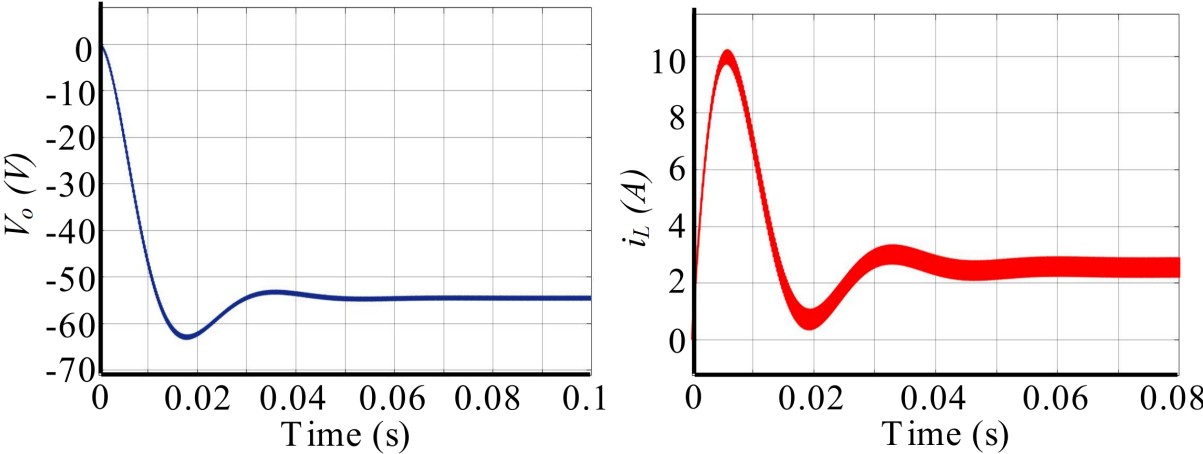

**Figure 12.** Analytical expression of capacitor voltage and inductor current over continous time from $S_4$.

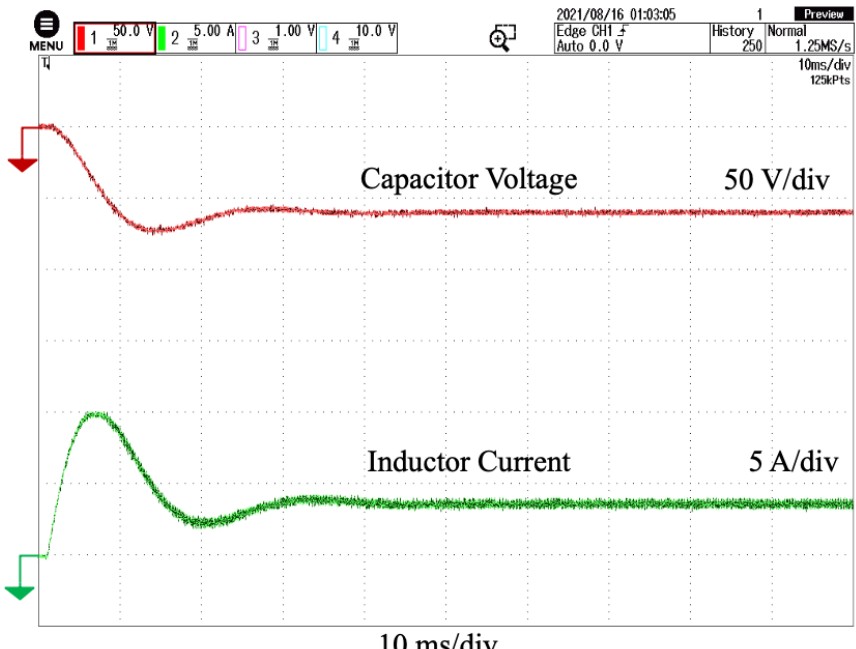

**Figure 13.** Experimental results of capacitor voltage and inductor current over continous time from $S_1$.

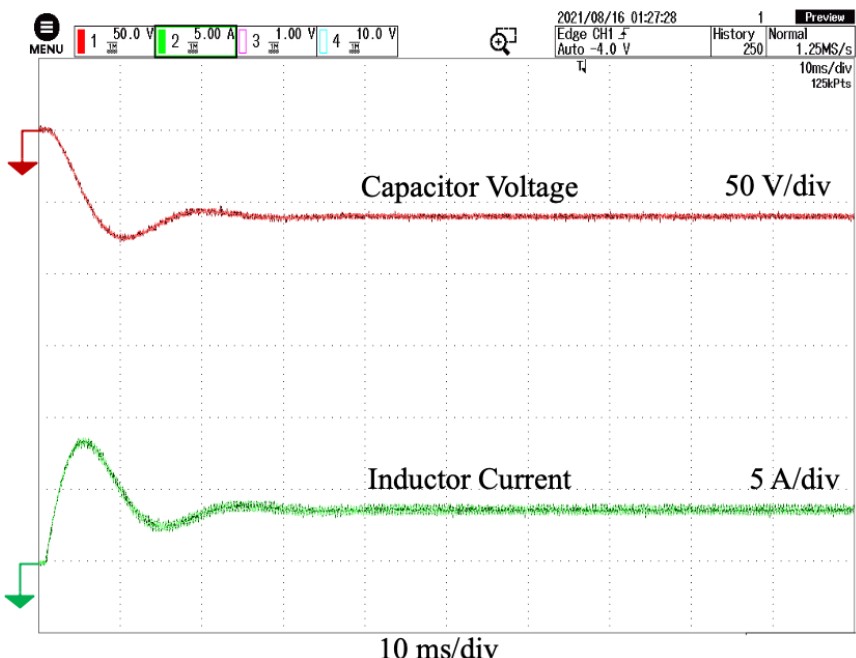

**Figure 14.** Experimental results of capacitor voltage and inductor current over continous time from $S_2$.

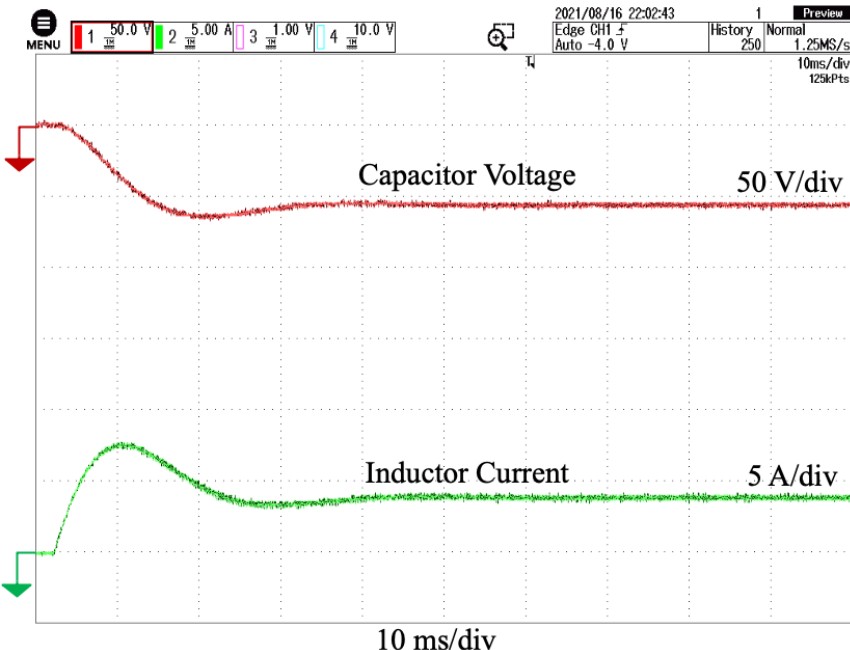

**Figure 15.** Experimental results of capacitor voltage and inductor current over continous time from $S_3$.

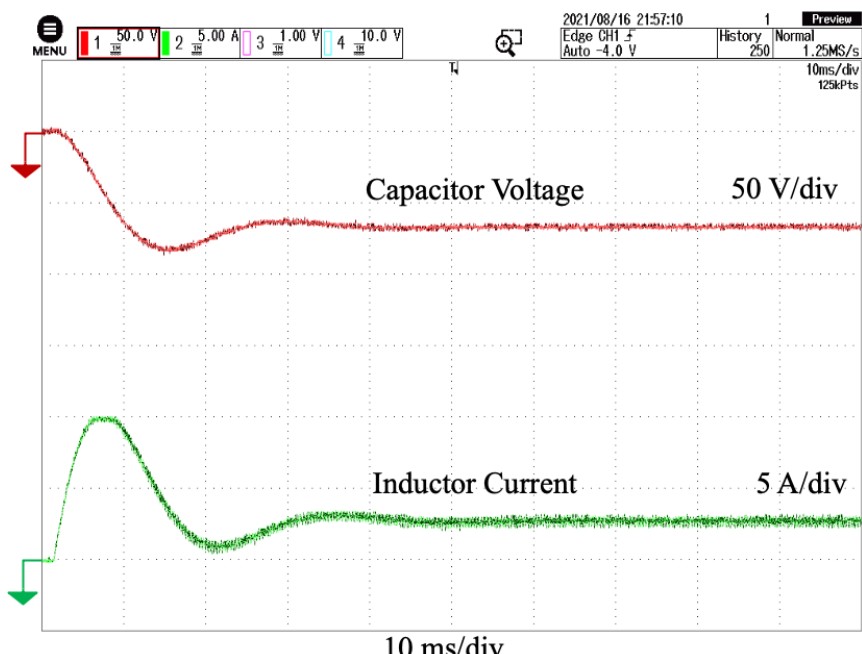

**Figure 16.** Experimental results of capacitor voltage and inductor current over continous time from $S_4$.

The results from the four parameter sets are investigated further to highlight several common transient parameters such as peak overshoot *P.O* and settling time $T_{ss}$. In general, the peak of overshoot *P.O* of a transient step response of a DC-DC converter is defined as

$$P.O = \frac{x_{max} - x_{ss}}{x_{ss}} \times 100\%,$$

where $x_{max}$ is the maximum value of the transient response and $x_{ss}$ is the steady-state response of the transient response. Settling time is defined here as the time it takes the transient response to reach steady-state. Table 2 provides the peak overshoot and settling time for each of the four parameter sets. Parameter set $S_1$ is taken to be the reference parameter set. As the capacitance of the buck-boost converter decreases from $C = 220 \times 10^{-6}$ to $C = 120 \times 10^{-6}$, i.e., parameter set $S_2$, there exists a decrease in both the settling times of $i_L$ and $v_C$. The peak overshoot of $i_L$ exhibits a 41.7% decrease and the peak overshoot of $v_C$ exhibits a 3.29% increase. As the inductance increases from parameter set $S_1$ from $L = 5 \times 10^{-3}$ H to $L = 9 \times 10^{-3}$ H and $r_L$ changes from $r_L = 0.8$ Ω to $r_L = 1.2$ Ω, i.e., parameter set $S_3$, the settling times of both $i_L$ and $v_C$ decrease. The peak overshoot of $i_L$ decreases by 57.5% and the peak overshoot of $v_C$ decreases by 5.29%. As the load resistance increases from $R = 60$ Ω to $R = 100$ Ω, i.e., parameter set $S_4$, the settling time of both $i_L$ and $v_C$ increase. The peak overshoot of $i_L$ exhibits a 138.3% increase and the peak overshoot of $v_C$ exhibits a 3.91% increase.

**Table 2.** Comparison of peak overshoot and settling time across the four different parameter sets.

| Parameter Set | *P.O* of $i_L$ [%] | $T_{ss}$ of $i_L$ [s] | *P.O* of $v_C$ [%] | $T_{ss}$ of $v_C$ [s] |
|:---:|:---:|:---:|:---:|:---:|
| $S_1$ | 162.1 | 0.056 | 21.38 | 0.055 |
| $S_2$ | 120.4 | 0.038 | 24.67 | 0.036 |
| $S_3$ | 104.6 | 0.052 | 16.09 | 0.049 |
| $S_4$ | 300.4 | 0.072 | 25.29 | 0.060 |

## 7. Conclusions

This paper proposes a new analytical modeling technique designed for power electronic DC-DC converter systems. The proposed technique allows for very accurate modeling of the transient behavior of DC-DC converters. Furthermore, unlike state-space averaging, which is the most commonly used method across the literature for DC-DC converter modeling, the proposed method can be applied to analyze the high-frequency components of DC-DC converters. The proposed analytical technique is a generalization of the analysis and modeling presented in [19–21]. First, the theory behind the methodology is presented thoroughly. In the circuit topology of a DC-DC converter, there exists a finite number of distinct circuit stages. Each of these circuit stages are linear. However, due to the switching elements in the circuit, the overall circuit topology is non-linear; rather, it is piecewise linear. Each circuit stage is treated as a continuous-time linear system. As such, each system can be solved through the uni-lateral Laplace transform in terms of the initial conditions of the inductor currents and capacitor voltages (ICCV). The initial conditions of each circuit stage at specific switching periods are related through discrete-time equations that originate the continuity of the ICCV with respect to time. These discrete equations are then solved through the Z-transform to determine the ICCV at the beginning of each circuit stage at each switching period. Next, the presented theory is applied to a non-ideal buck-boost converter circuitry for the purpose of clarification to the reader. Finally, the theory presented regarding the buck-boost converter is validated through hardware-in-loop (HIL) testing. The effects of the inductances, capacitances, and the PWM duty cycle on the ICCV are investigated. In the future, the authors plan on applying the proposed technique to higher order converters such as the SEPIC or the Ćuk converter.

**Author Contributions:** Conceptualization, A.M., M.S.B. and H.M.; methodology, A.M., M.S.B. and H.M.; software, A.M., M.S.B. and D.A.; validation, A.M., M.S.B. and D.A.; formal analysis, A.M., M.S.B. and H.M.; investigation, A.M., M.S.B. and D.A.; resources, A.M., M.S.B. and D.A.; data creation, A.M., M.S.B. and D.A.; writing—original draft preparation, A.M., M.S.B. and D.A.; writing—review and editing, A.M., M.S.B. and D.A.; visualization, A.M., M.S.B. and D.A.; supervision, M.S.B. and D.A.; project administration, M.S.B. and D.A.; funding acquisition, M.S.B., D.A. and H.M. All authors have read and agreed to the published version of the manuscript.

**Funding:** Princess Nourah bint Abdulrahman University Researchers Supporting Project number (PNURSP2022R137), Princess Nourah bint Abdulrahman University, Riyadh, Saudi Arabia. This work was supported by the research grants (SEED-2022-CE-103); Prince Sultan University; Saudi Arabia.

**Data Availability Statement:** Not applicable.

**Acknowledgments:** The authors would like to acknowledge the support of Prince Sultan University for paying the Article Processing Charges (APC) of this publication.

**Conflicts of Interest:** The authors declare no conflict of interest.

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
