# Peer review of "Analytical Solution for Transient Reactive Elements for DC-DC Converter Circuits"

_electronics, doi:10.3390/electronics11193121_

Round 1

Reviewer 1 Report

The paper lacks a review of the existing calculation methods for simulating such circuits. There is also no comparison of what the presented methods are better than the existing ones.

Reviewer 2 Report

The article covers a very interesting topic, however, its presentation needs some corrections:
1. Some of the charts lack axis descriptions.
2. The graphs presented were not discussed by the authors.
3. No detailed analysis of the results was made.
4. For comparison purposes, it would be appropriate to group the graphs with each other (put on one coordinate system). It is impossible to confirm consistency when the axes are not identical.
5. There is no description of what new the proposed algorithm has brought in comparison with others.
6. Conclusions are very general.
7. Quite poor literature review.

Reviewer 3 Report

1. There are references to papers 9-10 years of age [9-11].  A clarification of the importance of the topic and improvement in this field is needed at the beginning of the paper.

1. Numerous figures include empty square brackets. To be removed.

2. Fig. 12 to 15. A schematic view of the circuit indicating measurement points of current and voltage (green and red line) should be included. Why are these curves significant.

Reviewer 4 Report

This paper presents a new method for "Analytical Solution for Transient Reactive Elements for DC-DC Converter Circuits". In general, the paper is well written and the topic is interested. The overall quality of the paper is up to standard. However, in order to further enhance the quality of the paper, I have the following comments.

(1) The introduction in section 1.0 can be further extended to explain the relative merits of the proposed methods as compared to the methods used in past research. The comparison of the past research work with the proposed algorithms could be in form of a table.

(2)  A flowchart can be included in section 3 to summarize the step by step procedures of the proposed algorithms. 

(3) The conclusion in section 6 can be extended to describe more about the original contribution of the proposed methods.  

Round 2

Reviewer 2 Report

The authors have incorporated the suggestions indicated in the review. The revised article can be published.